# CryoEM structure and assembly mechanism of a bacterial virus genome gatekeeper

Igor Orlov[1,5,6], Stéphane Roche[2,6], Sandrine Brasilès[2], Natalya Lukoyanova [3], Marie-Christine Vaney[4], Paulo Tavares [2] ✉ & Elena V. Orlova [3] ✉

Numerous viruses package their dsDNA genome into preformed capsids through a portal gatekeeper that is subsequently closed. We report the structure of the DNA gatekeeper complex of bacteriophage SPP1 (gp6$_{12}$gp15$_{12}$gp16$_6$) in the post-DNA packaging state at 2.7 Å resolution obtained by single particle cryo-electron microscopy. Comparison of the native SPP1 complex with assembly-naïve structures of individual components uncovered the complex program of conformational changes leading to its assembly. After DNA packaging, gp15 binds via its C-terminus to the gp6 oligomer positioning gp15 subunits for oligomerization. Gp15 refolds its inner loops creating an intersubunit β-barrel that establishes different types of contacts with six gp16 subunits. Gp16 binding and oligomerization is accompanied by folding of helices that close the portal channel to keep the viral genome inside the capsid. This mechanism of assembly has broad functional and evolutionary implications for viruses of the prokaryotic tailed viruses-herpesviruses lineage.

Many double-stranded DNA (dsDNA) viruses have a specialized portal complex at one 5-fold vertex of their icosahedral capsid through which dsDNA enters and exits the virion. Tailed prokaryotic viruses and herpes viruses form a viral capsid precursor, termed procapsid, prior to DNA packaging. The procapsid contains a built-in portal protein complex[1–5] which serves as a docking platform for the terminase. The complex of the portal and terminase constitutes the viral genome packaging motor that translocates dsDNA into the procapsid. Tightly packed DNA (>400 mg/mL) exerts a pressure on the capsid that can reach ~60 atm[6,7]. Dissociation of the terminase from the motor is followed by closure of the portal vertex to prevent leakage of the packed viral genome[3]. Sealing of the portal channel is achieved by binding head completion proteins to assemble a complex named connector in tailed prokaryotic viruses, which consists of stacked cyclical oligomers creating a channel for DNA traffic[3,8–10]. Opening of the closed

connector is necessary for DNA entry in the tail tube and its subsequent delivery to the host cell at the beginning of infection.

CryoEM and X-ray structures were reported for a number of portal proteins[5,8,11–14], for the complex of P22 portal with the adaptor protein[15], and for some isolated proteins that participate in this process[3,16–19]. CryoEM studies provided more structural information on the portal complex assembled after DNA packaging[10] and when the tail is attached to the portal vertex[8,20–23]. Nonetheless, the molecular mechanism behind assembly of the DNA gatekeeper at the post-viral genome packaging state is still unknown.

In bacteriophage SPP1, the gp15 and gp16 head completion proteins bind sequentially to the dodecameric portal protein gp6 after DNA packaging[9,10,19]. The gp15 adaptor protein extends the portal tunnel that is fastened afterwards by binding the gp16 stopper protein. Here we report a cryoEM structure of the 902 kDa bacteriophage SPP1

[1]Centre for Integrative Biology (CBI), Department of Integrated Structural Biology, IGBMC, Université de Strasbourg, 67404 Illkirch, France. [2]Université Paris-Saclay, CEA, CNRS, Institute for Integrative Biology of the Cell (I2BC), 91198 Gif-sur-Yvette, France. [3]Institute of Structural and Molecular Biology, Department of Biological Sciences, Birkbeck College, Malet Street, London WC1E 7HX, UK. [4]Institut Pasteur, Université Paris Cité, CNRS UMR3569, Unité de Virologie Structurale, 75015 Paris, France. [5]Present address: University of Glasgow, Scottish Centre for Macromolecular Imaging, Sir Michael Stoker Building, 464 Bearsden Road, Glasgow G61 1QH Scotland, UK. [6]These authors contributed equally: Igor Orlov, Stéphane Roche.
✉e-mail: paulo.tavares@i2bc.paris-saclay.fr; e.orlova@bbk.ac.uk

connector in the post-DNA packaging state at a resolution of 2.7 Å. The connector was extracted from tailless DNA-filled capsids to avoid capsid protein subunits surrounding it at mismatched positions. This strategy allowed us to focus image processing on analysis of the connector complex and achieve a high-resolution structure. The de novo tracing of its 30 polypeptide chains within the cryoEM map enabled us to determine an atomic model of the entire connector complex (gp6$_{12}$gp15$_{12}$gp16$_6$). Structural comparison of the connector protein components with their assembly-naïve states in solution infers a sequential path of conformational changes and different types of interactions engaged during the assembly of the viral DNA gatekeeper. This structural study reveals also how symmetry transition takes place within the connector to match tail symmetry.

## Results

### Connector overall structure

The 902 kDa SPP1 connector complex was purified from disrupted tailless DNA-filled capsids[9]. The connector structure was determined by single particle analysis cryoEM at a resolution of 2.7 Å (at 0.5 threshold of FSC) (Fig. 1, see Methods and Supplementary Table 1). This resolution allowed unambiguous de novo tracing of the polypeptide chains of gp6, gp15 and gp16 within the cryoEM map of the complex (Supplementary Figs. 1 and 2). We have found that the connector comprises 12 subunits of both gp6 and gp15, but only 6 copies of gp16 (Fig. 1) in contrast to previous reports[10,19,20]. Gp15 and gp16 are monomeric before they assemble in the connector[24]. After DNA packaging, gp15 binds to the portal forming a dodecameric cyclical

oligomer. This assembly step occurs independently of gp16[20]. Closure of the connector complex requires binding of gp16 to gp15, leading to assembly of the gp16 6-mer (Fig. 1) that retains phage DNA inside the viral capsid[10,20]. These observations combined with the connector structure obtained (Fig. 1) reveal that connector assembling involves gp6-gp15, gp15-gp15, gp15-gp16, and gp16-gp16 interaction events.

### The gp6 portal protein fold at the post-DNA packaging state

The polypeptide chain of gp6 was traced in the cryoEM density map from residue 28 to 470 out of 503 residues (Fig. 2a). The overall gp6 fold in the connector is similar to the fold of gp6 in the 13-mer assembly-naïve form (PDB 2JES)[12]. Gp6 encompasses four major domains: the crown, wing, stem, and clip (Fig. 2a, b). It also features a tunnel loop that protrudes towards the interior of the portal central tunnel and a β-hairpin at the outer surface of the stem domain (Fig. 2a, b).

The gp6 12-mer within the connector and the assembly-naïve gp6 13-mer[12] are very similar in the inner part of the wing domain (residues 29–54, 88–169, 178–208, and 369–420, with RMSD between the Cα positions of 0.7 Å) and in the portal stem (residues 55–62, 81–87, 256–280, 327–346, RMSD 0.5 Å) (Fig. 2b). However, noticeable structural differences were found within the crown area, tunnel loops, β-hairpins, and clip domains (Fig. 2b). The tunnel loop forms a short helix in gp6 within the connector, while this helix is absent in the gp6 13-mer[12] (Fig. 2b). The low cryo-electron density of the tunnel loop in the connector gp6 suggests that this segment is rather flexible. The most significant difference between the connector gp6 and the gp6 13-mer is found in the clip domain that makes the most extensive intersubunit

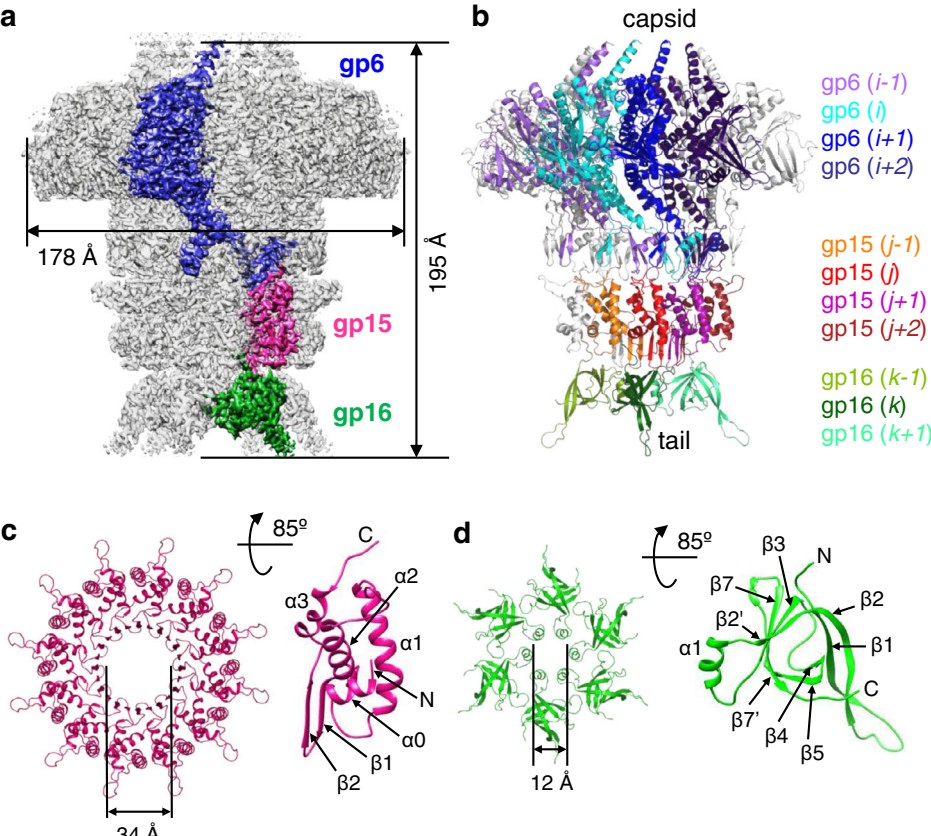

**Fig. 1 | Structure of the SPP1 connector (gp6$_{12}$gp15$_{12}$gp16$_6$) complex. a** CryoEM map of the SPP1 connector. One subunit of each gp6, gp15, and gp16 oligomer is shown in blue, magenta, and green, respectively. **b** Atomic model of the connector. For clarity, only the front half of the oligomer is shown. Four subunits of gp6 are displayed in purple, cyan, blue and dark blue. Four subunits of gp15 are in orange, red, magenta, and brown. Three copies of gp16 are in variations of green. Subunit

numbers with their colour codes indicated on the right are used consistently throughout the manuscript. **c** Gp15 dodecamer viewed from the capsid side (left panel). Secondary structural elements are labelled on one subunit (right panel). **d** Gp16 hexamer viewed from the capsid side (left panel). Secondary structural elements are labelled on one subunit (right panel).

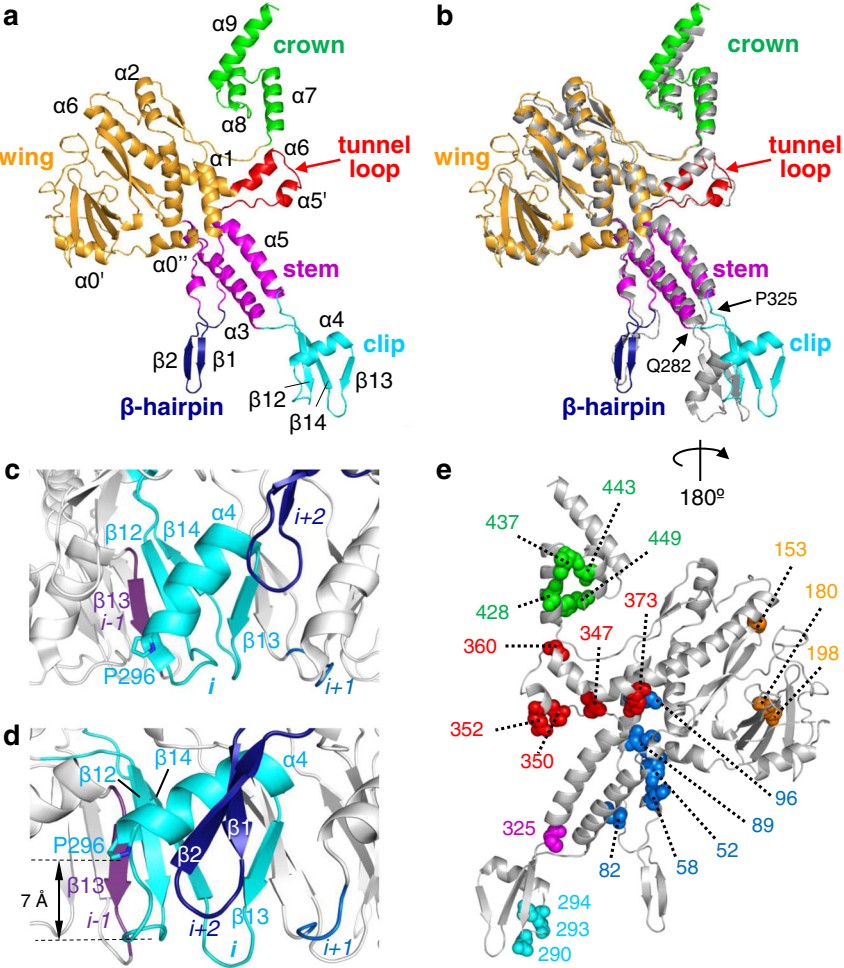

**Fig. 2 | Comparison of SPP1 portal protein gp6 structures and localization of mutations impairing viral DNA packaging. a** Secondary structural elements of the gp6 subunit in the SPP1 connector. Domains are colour coded: the crown domain (residues 421–470) is in green, the wing domain (residues 28–54, 88–255 and 369–420) in orange, the tunnel loop with the tilted region of helix α6 (residues 347–368) in red, the stem domain (residues 55–62, 81–87, 256–280 and 327–346) in magenta, the gp6 β-hairpin (residues 63–80) in blue and the clip domain (residues 281–326) in cyan. **b** Superposition of the gp6 subunit from the connector (12-mer) and the subunit of the gp6 assembly-naïve 13-mer (shown in grey; PDB 2JES). **c** Clip domain of the gp6 13-mer. **d** Clip domain of gp6 in the connector complex. Residues 307–312 of gp6, chain *i-1*, are shown in purple, 281–326 of chain *i* in cyan, residues 291–294 of chain *i + 1* in blue and residues 64–76 of chain *i + 2* in dark blue. The clip is in the same orientation as in **b**. 7 Å distance indicates the shift of α4 in the connector relative to α4 in the gp6 13-mer (see **c**). **e** Localization of mutations impairing SPP1 DNA packaging with residues coloured according to their mutation group (see Supplementary Table 2). The connector gp6 subunit is rotated 180° relative to **a** and **b**.

bonding in the two portal oligomers (Fig. 2c, d). The clip is formed by an intersubunit triple-stranded β-sheet, composed of strands β12 and β14 from subunit *i* and β13 from subunit *i-1*, and helix α4 from subunit *i* (Fig. 2c, d). These structural elements are found in both gp6 structures (12-mer and 13-mer) but exhibit visible shifts between residues Gln282 at the end of helix α3 and Pro325 at the beginning of helix α5 (Fig. 2b–d). In the gp6 13-mer, helix α4 has a tilted orientation[12] with its N-terminus Pro296 residue localized at the clip bottom (Fig. 2c; Supplementary Fig. 3a). In the connector gp6, Pro296 is shifted upwards by 7 Å bringing helix α4 to a more horizontal orientation closer to the gp6 β-hairpin of subunit *i + 2* (Fig. 2d). These changes in the clip domains of the connector gp6 lead to formation of pockets where the C-termini of gp15 bind to the portal oligomer (Supplementary Fig. 3b, c). Such pockets are absent in the assembly-naïve gp6 13-mer (Supplementary Fig. 3a) explaining why it does not bind gp15[24].

## Mapping of vital DNA packaging mutations in the portal structure

The structure of gp6 in the post-DNA packaging state allows to rationalize the effect of mutations previously shown to impair the SPP1 DNA

packaging process[25,26]. Since these mutated gp6 proteins are incorporated in procapsids, this implies that they are folded correctly and interact with the other procapsid components ensuring proper procapsid assembly[25]. Residues critical for DNA packaging[25,26] are located in different structural elements of the portal structure (Fig. 2e).

Several mutations mapped in the portal wing (Fig. 2e, in orange) apparently cause local destabilization of this domain (Supplementary Table 2). It is difficult to interpret how their effect on the structure disrupts DNA packaging through the portal central tunnel.

A set of mutations that block DNA packaging was found in the gp6 crown. This region forms the upper part of the tunnel channel through which DNA is translocated into the capsid interior. Mutations affecting the crown structural organization involve residues engaged in intersubunit bonding (Ser428, Ile437, Phe449) and/or destabilize the crown hydrophobic core (Ile437Val, Ala443Thr, Phe449Leu) formed by helices α7 and α8 (Fig. 2e, in green; Supplementary Table 2).

Another cluster of mutated residues affect either the gp6 tunnel loop structure (Ser350, Glu352), or the loop interaction with α5 (Val347, Lys373) and the portal crown (Gly360). They highlight a role of the loops in the DNA packaging process (Fig. 2e, in red;

Supplementary Table 2). We previously showed that substitution Glu352Gly leads to reduction of the terminase ATPase activity[27]. This cross-talk between tunnel loops and the terminase during DNA trans-location is mediated, conceivably, through helix α5 and the gp6 clip region[28].

DNA packaging is also disrupted by mutation of Pro325 (Fig. 2e, in purple) located at a hinge position between helix α5 and the clip. The polypeptide chains of the gp6 13-mer and of the gp6 connector 12-mer change their conformation at this point leading to a different position of the clip in the two structures (Fig. 2b–e). Substitution Pro325Leu disrupts inter-subunit bonding and might extend the length of helix α5, affecting the correct positioning of the clip elements.

The next important set of mutations is found in loops anchoring the gp6 β-hairpin at proximity to the wing and stem (Fig. 2b, in blue). We hypothesize that the mutations in this hairpin change its position with respect of the clip domain (Fig. 2d, β1-β2 in blue), preventing formation of the clip pocket. The essential role of the clip pocket in DNA packaging is revealed by the highly detrimental effect of amino acid substitutions in Asn290, Gly293, and Glu294[25] of loop β12-α4 (Fig. 2e, in cyan; Supplementary Table 2). These three residues are exposed to the portal exterior (Fig. 2e). Asn290 is located at the tip of loop β12-α4, with its side chain engaged in intrasubunit and inter-subunit hydrogen bonds with the carbonyl of Asp314 and with the nitrogen of Gly313, respectively (Supplementary Fig. 4). These inter-actions mimic beta interactions and are essential for the correct positioning of loops β12-α4 and β13-β14 within the clip domain. Sub-stitution of Asn290 disrupts one or both hydrogen bonds destabilizing these loops. Gly293 and Glu294 make no side chain contacts sug-gesting that the phenotype of their substitution results from a defect in the direct interaction with the terminase[29]. Disruption of this interac-tion upon disassembly of the packaging motor renders the gp6 clip pocket ready for gp15 binding. Sequential interaction of the terminase and gp15 with the same gp6 region of the clip is corroborated by finding that mutation Glu294Lys disrupts SPP1 DNA packaging while substitution of the same residue Glu294Gly impairs only the sub-sequent assembly step of binding to gp15[26] (Supplementary Table 2; see below). Collectively this structure-function analysis highlights structural elements lining the portal channel and the clip as essential factors for phage DNA packaging.

**The gp15 structure and gp6-gp15 interface.** Termination of DNA packaging is followed by departure of the terminase and attachment of gp15 to gp6, initiating connector assembly. The core of gp15 comprises 4 helices (Fig. 1c, right panel). Its C-terminus is directed towards gp6 while the hairpin β1-β2 points towards gp16. Gp15 interacts with the gp6 oligomer mostly via insertion of its C-terminus into the gp6 clip pocket (Fig. 3a, b; Supplementary Fig. 3c). The gp15 C-terminus of chain *j + 1* establishes nine interchain bonds with the clip of gp6 chain *i* (Supplementary Table 3). Gp15 His37 of chain *j+1* makes also a bond with clip residue Asp292 of gp6 chain *i+1* (Fig. 3b) while gp15 Arg102 establishes a hydrogen bond with the carbonyl of Asp68 from the β-hairpin of gp6 chain *i+2* (Fig. 3b; Supplementary Table 3). This inter-action network explains why mutation Glu294Gly in gp6, which dis-rupts its hydrogen bond with gp15 Met100 (Fig. 3b), or replacement of the five C-terminal residues (ArgLysMetAlaArg) in gp15 with MetAla-Gly, prevents gp15 stable binding to gp6[20,26]. These interactions are characterized by electrostatic potential complementarity between the gp15 C-terminus and the gp6 pocket and their shape matching (Sup-plementary Fig. 5a).

**Gp15 conformational changes**
Assembly-naïve gp15 (PDB 2KBZ) is a monomer in solution composed of an α-helical core, flexible loop α1-α2, a large unstructured loop between helices α2 and α3, and the flexible C terminus, as shown by NMR structures[19]. The α-helical core α1-α3 maintains its conformation

after assembly of gp15 in the connector but α0, loop α2-α3, and the C-terminus undergo major conformational changes (Fig. 3c, in green, blue, and orange, respectively). The gp15 C-terminus interacts with gp6 (Fig. 3a, b; Supplementary Fig. 3c). The N-terminus and α0 undergo 145° rotation to establish contacts with helices α1 and α2 (green arrow in Fig. 3c, right panel). Helix α0 of subunit *j* makes also links with the adjacent subunit through residues Arg5 and Arg8 forming salt bridges with Glu23 of α1 from subunit *j-1* (Fig. 3d). The most dramatic conformational change takes place by folding of loop α2-α3 into the β1-β2 hairpin that forms the gp15 24-stranded inter-molecular β-barrel (Figs. 1b, c, and 3c).

**Gp15 intersubunit interactions within the connector**
The surface of interaction between a single gp15 subunit and gp6 in the connector is only ~700 Å² and the calculated dissociation energy is −1.6 kcal/mol as assessed by EBI-PISA[30] (Supplementary Table 4). This indicates that the single gp15 monomer does not bind strongly to the gp6 ring[31]. In contrast, adjacent gp15 subunits have an interaction surface of ~1250 Å² and a dissociation energy of −13.3 kcal/mol (Sup-plementary Table 4). Furthermore, each gp15 subunit establishes interfaces with two adjacent gp15 subunits as its cyclical 12mer forms during connector assembly. The initial attachment of gp15 monomers to gp6 is thus followed by gp15 intersubunit interactions that ensure stable association of gp15 to the gp6 oligomer.

The gp15-gp15 interface comprises lateral contacts between helix α1 of subunit *j-1* and helices α0 and α2 of the adjacent subunit *j* (Supplementary Table 3). Furthermore, helix α0 binds helix α1 of adjacent subunits at the connector periphery through a network of salt bridges and hydrogen bonds (Fig. 3d). The gp15 β1-β2 hairpins of neighbour subunits form an intersubunit 24-stranded β-barrel distal from the portal (Fig. 1b, c) that outlines the 34 Å-wide central tunnel of gp15 (Fig. 1c). Within each gp15 subunit the β-hairpin is connected to its α-helical core via loops α2-β1 and β2-α3 (Fig. 3c, e). The carboxylate group of Glu85 in loop β2-α3 of subunit *j* establishes five hydrogen bonds with the main chain and side chains of Ser88 and Thr89 of subunit *j + 1* (Fig. 3e), stabilising the overall positioning of the β-barrel strands.

**Gp16 structure and the gp15-gp16 interface**
The bottom part of the connector is formed by six gp16 subunits. Each of them comprises a β-barrel core, the tunnel α-helix α1, and the β1-β2 loop that protrudes below the connector and presumably interacts with the tail (Fig. 1d).

The interface between the gp15 dodecamer and the gp16 hexamer is characterized as well by electrostatic complementarity (Supple-mentary Fig. 5b). Most of the gp15-gp16 interactions involve the bot-tom rim of the gp15 24-strand β-barrel (Figs. 1b and 3f, g). Each gp16 subunit makes contacts with four gp15 subunits (Fig. 3f; Sup-plementary Table 3). Remarkably, residues Thr77 of three different gp15 subunits interact with the same gp16 subunit *k* (Fig. 3f). Thr77 of gp15 chain *j + 1* makes a hydrogen bond with gp16 Ser91 while Thr77 of gp15 chain *j* establishes hydrogen bonds with residues Tyr61 and Arg98 of gp16. Finally, Thr77 of gp15 of chain *j-1* is involved in two hydrogen bonds with gp16 Gln39. Additionally, Arg73 of gp15 chain *j* makes a salt bridge with Glu95 of gp16 (Fig. 3f). Disruption of this interaction by mutation Arg73Glu in gp15 was previously shown to prevent stable binding of gp16 to gp15[20]. A second type of interactions between gp16 and gp15 takes place by insertion of the gp16 N-terminus from chain *k + 1* in a crease between the gp15 β-barrel and loops α2-β1 of gp15 chains *j* and *j + 1*, respectively (Fig. 3g). This network of interactions establishes the transition of 12 to 6-fold symmetry within the con-nector. Overall, the interaction between each subunit of gp16 and the gp15 oligomer has a surface of 1040 Å² and a dissociation energy of −11.3 kcal/mol (Supplementary Table 4) indicating on strong bonding between gp15 and gp16 oligomers.

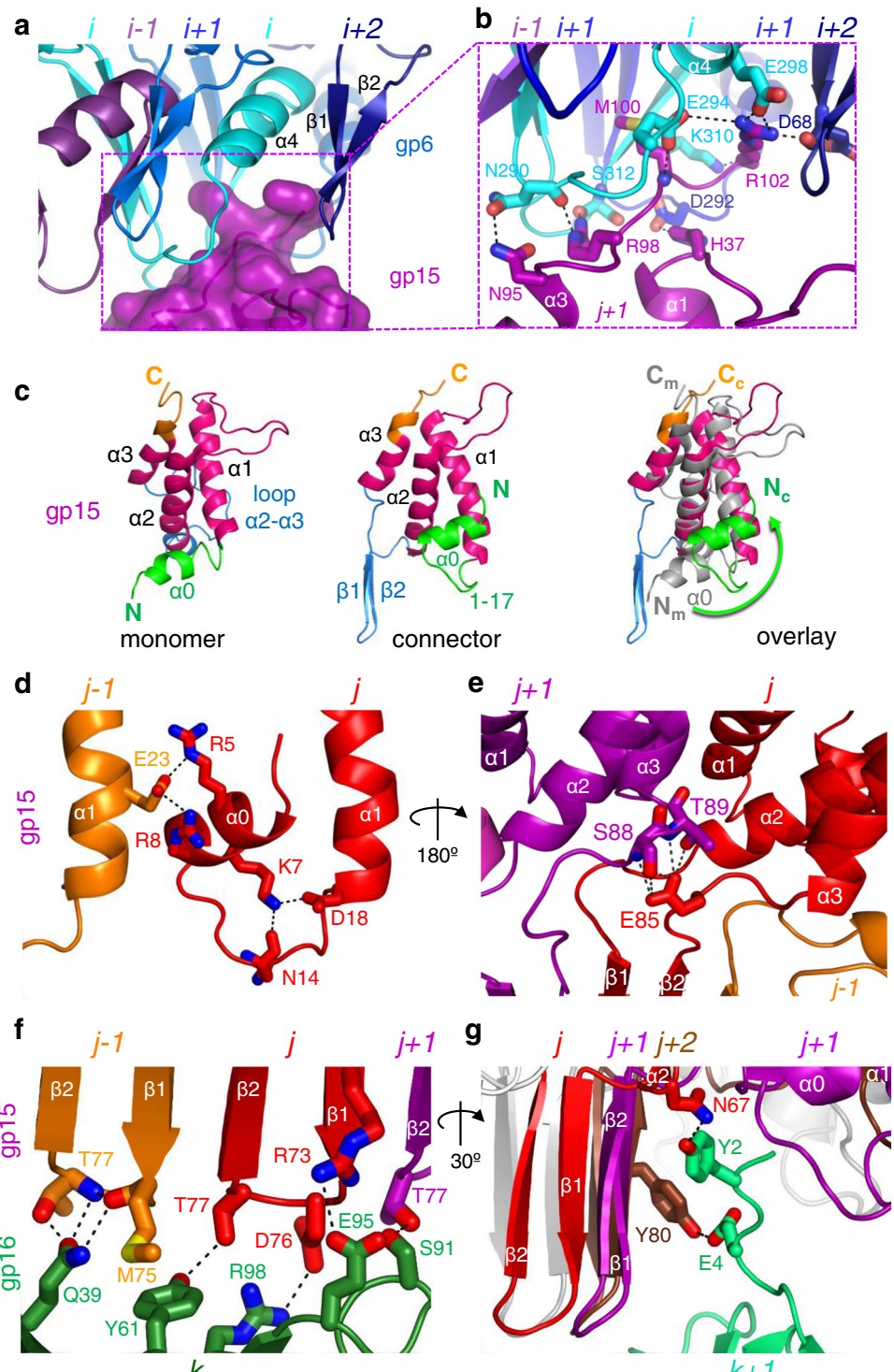

**Fig. 3 | Intermolecular interfaces within the SPP1 connector. a** gp6-gp15 inter-face. The C-terminus of the gp15 subunit *j + 1* (in magenta) interacts with the clip pocket formed by three gp6 subunits shown in cyan (*i*), blue (*i + 1*) and dark blue (*i + 2*) as in Fig. 1b. View from the outside of the connector. **b** Zoomed in view of the gp6-gp15 interactions of the outlined area in **a**. Interchain hydrogen bonds and salt bridges between gp6 and gp15 subunits are displayed as dashed lines in black. **c** Comparison of the assembly-naïve gp15 monomer (left) (PDB 2KBZ) and the gp15 subunit in the connector (centre) superposed through their α-helical core (magenta). Helix α0 is shown in green, loop α2-α3 is in blue, and the C-terminus is in orange. The overlay of the gp15 subunit from the connector with the NMR

monomer (in grey) is shown in the right panel. The green arrow indicates rotation of helix α0 on -145°. The N and C-termini of the gp15 NMR monomer (N$_m$ and C$_m$) and of the gp15 subunit in the connector (N$_c$ and C$_c$) are labelled in the overlay. **d** Intersubunit interaction of gp15 α0 with helix α1 of adjacent subunit viewed from outside of the connector. **e** Gp15 Glu85 intersubunit bonding viewed from the connector tunnel. **f** View from the outer surface of the β-barrel shows the interface contacts between the gp15 β-barrel bottom rim and gp16. Dashed lines indicate hydrogen bonds and salt bridges between one gp16 subunit and three gp15 subunits. **g** Side view of the β-barrel showing the interaction between gp15 and the gp16 N-terminus.

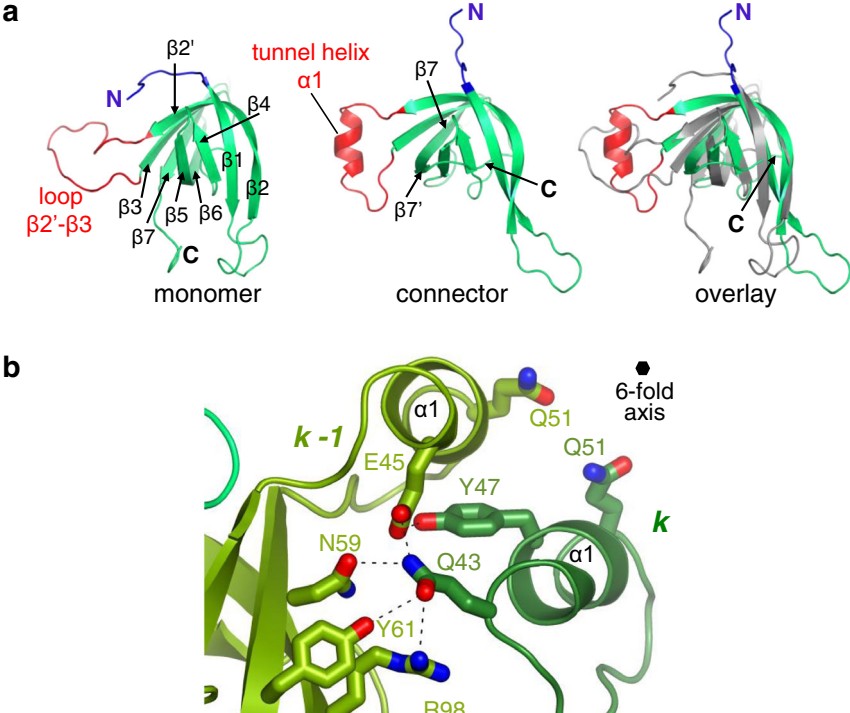

**Fig. 4 | Conformational changes of gp16. a** Comparison of the NMR structure of the assembly-naïve gp16 monomer (left panel, PDB 2KCA) with one subunit of gp16 in the connector (central panel). The N-terminus is shown in dark blue (residues 1–7) and loop β2'-β3 is in red (residues 40–56). The right panel shows the overlay of the two gp16 structures aligned through their β-sheet core. The NMR structure is in grey. **b** Tunnel helices of two adjacent subunits of the gp16 hexamer viewed from the capsid side. Black dashed lines show hydrogen bonds. The position of the 6-fold symmetry axis is shown by a black hexamer.

## Gp16 conformational changes and portal tunnel closure

The gp16 subunit in the connector has a well-defined eight-stranded β-barrel core (Fig. 1d) like the assembly-naïve gp16 (PDB 2KCA) monomer in solution (Fig. 4a). The most dramatic change within the gp16 structure takes place in flexible loop β2'-β3 (residues 43–51) that folds into the tunnel α-helix (Fig. 4a). The α-helices of six gp16 subunits in the connector are held together by 5 intersubunit hydrogen bonds between adjacent subunits. Gln43 makes four of them and the fifth is between Tyr47-Glu45 (Fig. 4b). The tunnel helices form a constriction in the portal channel with a diameter of ~12 Å between side chains of opposite gp16 Gln51 residues (Fig. 1d). This is the only region of the connector tunnel narrower than the 20 Å diameter of a DNA double-helix, locking DNA inside the capsid after its encapsidation. The interaction between gp16 subunits is notably weak with a surface of 510 Å² between adjacent subunits and a calculated dissociation energy of −2 kcal/mol (Supplementary Table 4), suggesting that bonds to gp15 subunits play a significant role to hold the gp16 hexamer in the connector. The gp16 N-terminus (residues 12–30) forms loop β1-β2, that protrudes below the connector and presumably interacts with the tail (Fig. 1d). Tail binding to gp16 might further stabilise its structure in the infectious tailed phage[20].

## Discussion

The 2.7 Å resolution structure of the SPP1 gp6₁₂gp15₁₂gp16₆ complex reported here uncovers the detailed molecular architecture of the complete viral DNA gatekeeper from tailed prokaryotic viruses. Comparison with the structure of its non-assembled components reveals the molecular events leading to assembly of the gatekeeper (Fig. 5) that have extensive functional and evolutionary implications.

The connector structure shows the conformation of the portal protein gp6 in its post-DNA packaging state. Locations of mutations that arrest DNA packaging before the endonucleolytic cleavage that terminates the process[25,26] identify clusters of functionally important residues within structural elements of gp6 (Fig. 2e). Two clusters were found in the crown and in loops that line the portal tunnel. Other mutations affect structural elements that build the clip pocket exposed for binding of the terminase (Fig. 2e and Supplementary Fig. 3b). The previous finding that motion of stem helix α5 is necessary for DNA packaging[28] and localisation of mutations in the gp6 structure suggest that the tunnel loops act in a concerted mode with the terminase during DNA packaging[27,29]. Their cross-talk through helix α5 might support DNA translocation, driven primarily by the terminase pumping activity[29], and a ratchet mechanism in which tunnel loops act to prevent sliding of DNA out of the phage capsid as it reaches high concentration inside the capsid. The previously proposed ratchet function of the tunnel loops[8,32,33], is conceivably coordinated with the DNA packaging motor activity. This retention mechanism operating during genome packaging is transient. Keeping the DNA inside the capsid requires binding of gp15 and gp16 to the portal system upon terminase departure[10]. The process is coordinated by sequential interaction of the terminase and gp15 with a common binding interface within a pocket in the gp6 clip (Figs. 2e and 3a, b).

Comparison of the NMR structures of SPP1 assembly-naive gp15 and gp16[19] with their conformations within the connector highlights the structural rearrangements occurring during assembly of the DNA gatekeeper (Figs. 3c, 4a, and 5). The analysis of gp15 and gp16 structural homologues (Supplementary Figs. 6–8) expands further our understanding of their conformational landscape. Gp15-like proteins have a conserved α-helical core but differ in the conformations of loop α0-α1, loop α2-α3, and/or of the C-terminus. These variations inform on structural changes on the pathway from the monomer to the dodecameric state (Supplementary Fig. 6). The C-terminus of SPP1 gp15 has a defined conformation only if it is bound to the portal protein. That corroborates with its role in anchoring the gp15-like adaptor proteins to the portal clip (Fig. 5a). During connector assembly, gp15

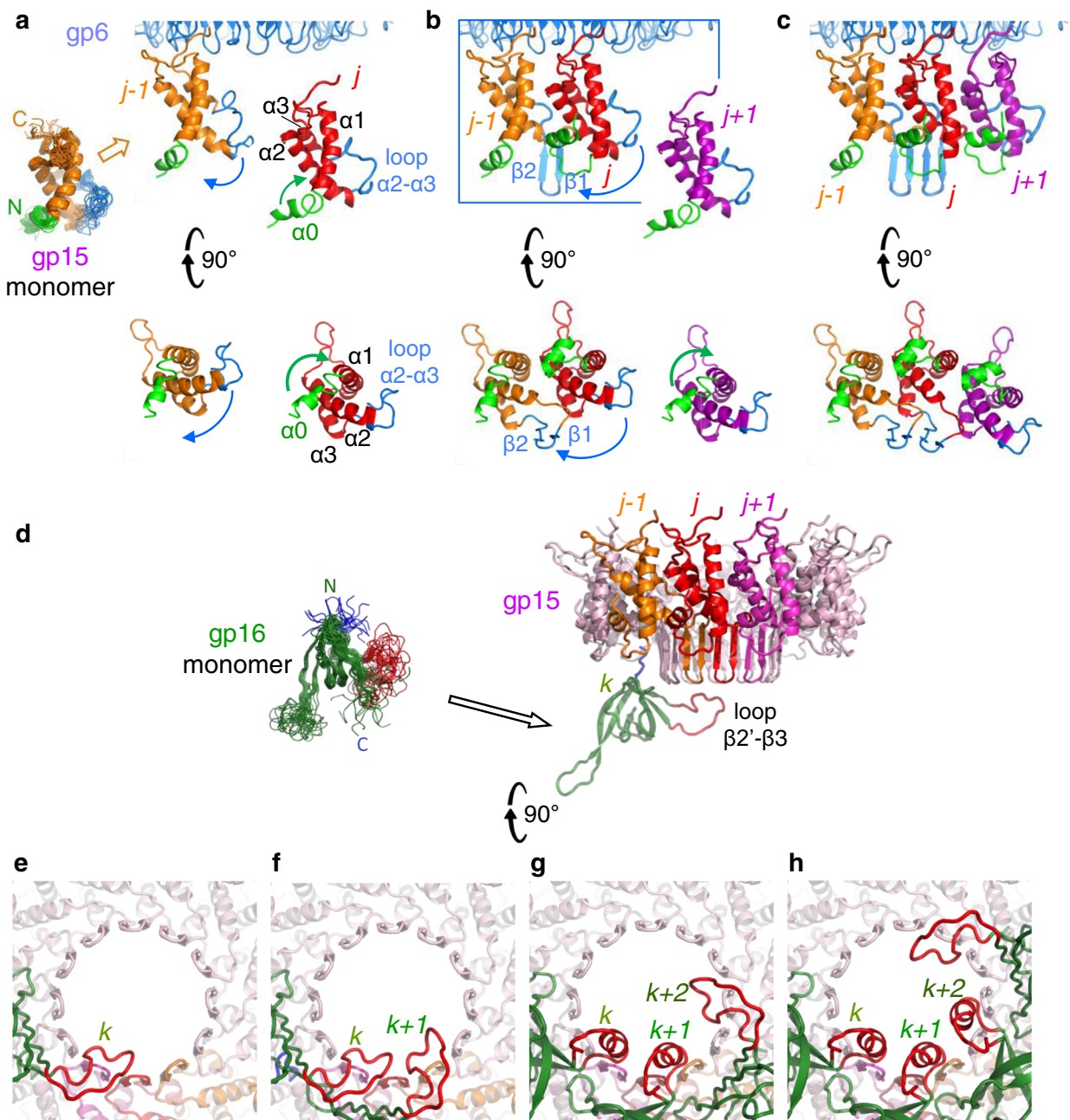

**Fig. 5 | Model of connector assembly. a** Anchoring of the gp15 monomer in the gp6 clip. Twenty models of the gp15 monomer NMR structures[19] are superposed to highlight stable and flexible structural elements before assembly (left panel). The N-terminus α0 and loop α2-α3 (shown in green and blue, respectively) undergo conformational changes during assembly. **a–c** Binding to gp6 of adjacent gp15 subunits leads to clashes between gp15 loops α2-α3 inducing changes and folding of the β1-β2 hairpin (bent blue arrow in **a** indicates the change occurring in **b** upon oligomerization), building the gp15 intersubunit β-barrel. Helices α0 reposition during oligomerization (bent green arrow in **a** shows the change occurring in **b**), to bridge helices α1 of adjacent subunits. Views from the tail side in the bottom part of the figure show only gp15 subunits for clarity. **d** Binding of gp16 to the gp15 ring. Superposed twenty models of the gp16 monomer NMR structure[19] are shown on the left panel. The gp16 N-terminus and loop β2'-β3, that change conformation during assembly, are rendered in dark blue and red, respectively. **e-h** Folding of gp16 loops β2'-β3 to tunnel helices stabilized by inter-helices interactions following oligomerization of gp16 in the connector. Views are shown from the phage tail side.

helix α0 changes its position to a more horizontal orientation (Figs. 3c and 5a, b). Interestingly, in the non-assembled monomer of the gp15 homologue YqbG[34] helix α0 adopts a position, stabilized by intrachain bonds, that is close to the orientation found in the gp15 connector state (Supplementary Fig. 6b). In SPP1, such conformation requires intersubunit bonding of α0 that bridges helices α1 of neighbour subunits (Fig. 3d). Disruption of these interactions in mutant gp15 Arg5Glu Arg8Glu (Fig. 3d) allows for formation of the gp15-g16 complex in phage capsids but prevents closure of the connector tunnel[20]. Re-

positioning of α0 is thus an essential assembly step (steps **a** to **b** in Fig. 5; Supplementary Movie 1).

The NMR structures of gp15 and YqbG monomers[19,34] both show highly mobile loops α2-α3 (Supplementary Fig. 6b). Their α-helical cores have similar intrasubunit interactions. The homologue gp6 protein from siphophage HK97 forms a 13-mer ring in assembly-naïve conditions[18] and the loop α2-α3 of HK97gp6[18] has an antiparallel β-like interchain interaction (Supplementary Figs. 6b and 7a). This loop is located in a position similar to loop α2-α3 from SPP1 gp15 that folds to build a β-

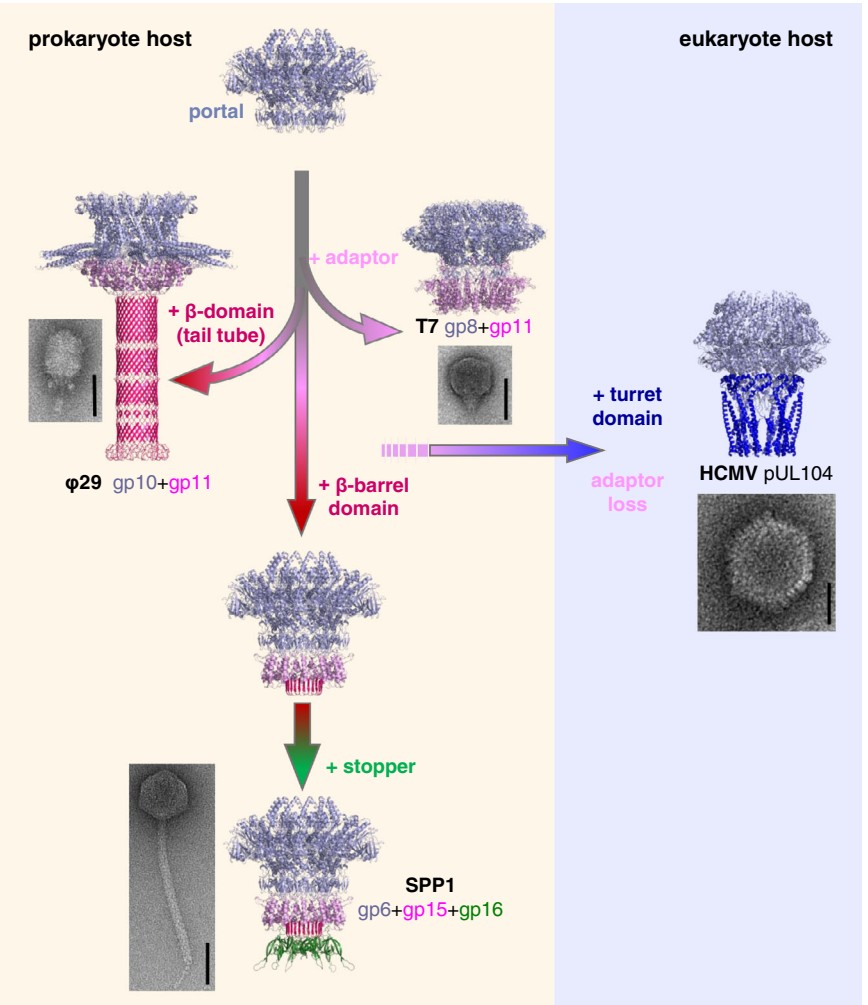

**Fig. 6 | Evolutionary divergence of the prokaryotic tailed viruses-herpesviruses lineage at the portal vertex.** The portal protein (in grey-blue) is conserved in all viruses of the lineage. The adaptor protein (light magenta) that binds to the portal clip of prokaryotic viruses has various structural features in the region distal from the portal. They define two evolutionary division points for viruses with short tails, typified by phages T7 and φ29, from the phages with long tails. Adaptor proteins of phages with long tails encompass a β-barrel (magenta) for binding stopper proteins (in green). Addition of a turret domain (in blue) to the portal and loss of the adaptor protein in herpesviruses marks their divergence from prokaryote viruses. Domains of the herpes cytomegalovirus (HCMV) portal protein and of tailed phage adaptors that define breakpoints in the lineage evolutionary trajectory are shown in dark blue and magenta, respectively. The structures presented are the SPP1 connector (gp6$_{12}$gp15$_{12}$gp16$_6$; PDB 7Z4W this work), HCMV pUL104$_{12}$ (PDB 7ETM), T7 gp8$_{12}$gp11$_{12}$ (PDB 6R21), and φ29 gp10$_{12}$gp11$_{12}$ (PDB 6QZF). Representative images of a herpesvirus capsid and mature phage particles in negative stain (2% uranyl acetate) are shown close to the corresponding connectors. The scale bar corresponds to 50 nm.

sheet within the connector (Supplementary Figs. 6b and 7b). Although gp6 of HK97 forms most likely a 12-mer when it assembles in the phage particle[18], the 13-mer structure suggests that its oligomerization leads to formation of β-like interactions between the α2-α3 loops of adjacent subunits. We hypothesize that they represent a precursor state of an inter-subunit β-barrel in HK97, as was found in SPP1 gp15. Rigid-body fitting of gp15 NMR monomers into the SPP1 connector exposes multiple clashes between loops α2-α3 of adjacent subunits, dictating conformational changes in their fold. These changes result conceivably in formation of the β-barrel during gp15 oligomerization (**a** to **c** in Fig. 5; Supplementary Movie 1). Collectively, analysis of SPP1 gp15 homologous protein structures reveal different conformations that are necessary for achieving the functional gp15 assembled state.

The gp15 β-barrel creates a platform for gp16 attachment to the SPP1 portal complex (Fig. 5d). Connector stoppers, typified by gp16, are characterised by a common β-barrel core[3] (Supplementary Fig. 8). The monomers of phage λ gpFII[17] (PDB 1K0H) and XkdH of prophage PBSX (PDB 3F3B) have a flexible loop equivalent to the SPP1 gp16 monomer β2'-β3 loop. This loop is folded into a α-helix in the

monomer of the putative stopper SF1141 (Supplementary Fig. 8b). Loop β2'-β3 of SPP1 gp16 folds also into a α-helix after binding of gp16 subunits to the gp15 ring in SPP1 (step f to h in Fig. 5; Supplementary Movie 1). Five inter-helices contacts stabilize the helical conformation occluding the portal channel (Fig. 4b) while not imposing a strong barrier for disruption of this bonding that is necessary for subsequent channel opening. The weak gp16 intersubunit bonding (Supplementary Table 4) indicates that its interaction with gp15 plays a major role for endorsed positioning of the six gp16 subunits in the connector. This corroborates with mutations in gp15 that impair specifically portal tunnel closure by gp16[20] (see above). The 12 to 6-fold symmetry transition within the gp15-gp16 complex enables a direct match of the connector with the structure of the phage pre-assembled long tail that has 6-fold rotational symmetry.

The portal vertex complex represents an essential DNA traffic gate conserved among all viruses of the prokaryotic-tailed viruses-herpesviruses lineage[35]. The portal protein component is conserved in the lineage. In contrast, the effectors acting after DNA packaging diverged as viruses evolved to infect different cellular hosts (Fig. 6). Eukaryotic

herpesviruses furnished the portal protein clip with "tentacle" α-helices that protrude towards a cap that closes the portal system[4,36] (Fig. 6). Clip domains, like in SPP1 gp6, are one of the most conserved structural elements in the portal protein from tailed bacteriophages[3,14]. These viruses have also gp15-like proteins that form dodecamers extending the portal DNA tunnel. However, the β-barrel region of SPP1 gp15 has changed significantly in the homologous proteins from phages with short tails (Fig. 6). Phages P68 and φ29 have a β-barrel longer than 100 Å forming the short tail DNA conduit[23,37] (Supplementary Fig. 7c, d) while T7[8] and possibly P22[15] have no equivalent to the SPP1 gp15 β-barrel (Fig. 6). This evolutionary divergence is further marked by the absence of gp16 homologous proteins in phages with short tails, with the exception of a domain from the phage φ29 tail knob protein gp9[23] (Supplementary Fig. 8). In contrast, gp16-like proteins are conserved among phages with long tails[38]. The molecular architecture of the gp15-gp16 interface accomplishes the connector-to-tail symmetry transition and matches the connector tunnel-forming gp15 structure with the tail tube-like fold of gp16[39]. This β-barrel fold is the common feature of some components of long tail tubes[39,40]. The gp16-like proteins evolved to establish three main functions: firstly the interaction with gp15-like proteins, secondly to build a temporary retention system of the viral DNA, and thirdly to bind to the tail-to-head joining protein at the tail tube end[3,20,21,41,42]. The high resolution of the connector multi-protein complex enables us to reveal the most important residues involved into interactions between protein components, how transition between different symmetries takes place, and which protein components are responsible for regulation of DNA transfer through the main channel of the system.

The architecture of the connector and its assembly mechanism reveal the evolutionary process of the tailed prokaryotic viruses-herpesviruses lineage (Fig. 6). In tailed prokaryotic viruses infecting bacteria and archaea there are gp15-like proteins that bind to the portal clip[38,43]. Their structures reveal branching points in the lineage to form short tails or to connect the gp16-like proteins (Fig. 6). The latter interaction creates an interface for attaching long tail tubes assembled in an independent pathway. Prokaryotes preceded eukaryotes in evolution. We thus hypothesize that an ancestral prokaryotic-tailed virus evolved by insertion of the turret domain in the portal clip and loss of interaction with the adaptor protein to originate the portal ancestor of eukaryotic herpesviruses virions (Fig. 6). Collectively, our connector structure and comparative structural biology study uncover the molecular mechanism of assembly of the complete viral DNA gatekeeper, providing a long-waited molecular framework to trace the evolutionary divergence path between herpesviruses and different tailed phage families.

## Methods

### Connector complex purification

The SPP1 connector complex purification was reported[9]. Briefly, SPP1 tailless particles were produced by infection of the non-permissive host *Bacillus subtilis* YB886 with phage SPP1*sus9*[42,44] and purified by isopycnic centrifugation in a discontinuous CsCl gradient[45]. Tailless particles were disrupted with 50 mM EDTA for 30 min at 55 °C. Then, MgCl₂ was added to a final concentration of 100 mM and free DNA was digested overnight with 50 U Benzonase in an oven at 37 °C. Connector complexes were then purified by sedimentation through a 10–30% (w/v) glycerol gradient in TBT buffer (100 mM Tris-Cl, pH 7.5, 100 mM NaCl, 10 mM MgCl₂) run at 35,000 rpm in a SW41 rotor (Beckman Coulter) for 3 h at 4 °C. Fractions containing connectors were identified by the presence of gp6, gp15 and gp16 in SDS-PAGE and western blots[9]. They were then pooled followed by concentration and buffer exchanged to 0.5x TBT in a Vivaspin micro concentrator with a cut-off of 100 kDa.

### Cryo-EM grid preparation and data acquisition

3 μL of suspension of purified gp6gp15gp16 connector complexes at 0.64 mg/mL were applied to a freshly glow-discharged (ising PELCO Easiglow,Ted Pella, USA) C-flat grid (Protochips, USA; 2/2 400 mesh) and blotted for 5 seconds before plunge-freezing the grids and vitrified using a Vitrobot Mark IV (ThermoFisher™) at 96% humidity, and 4 °C. Data were collected on a Titan Krios microscope (ThermoFisher™, the eBIC Diamond light source facility, Harwell, Oxfordshire, UK) operating at 300 kV with the specimen maintained at liquid nitrogen temperatures. Images were recorded using a Falcon III camera (ThermoFisher™) in integrating mode that enables faster data acquisition. Data collection was done using the EPU software (ThermoFisher™) within a defocus range of −1.2 to −2.5 μm. Movies (40 frames per movie) were collected with a dose of 1.12 e⁻/Å² per frame at the specimen plane over a 1 s exposure. The calibrated pixel size was 0.69 Å at the specimen level.

### Image processing

3876 movies were aligned using MotionCorr2[46]. CTFFIND4[47] was used to determine defocus values (Supplementary Fig. 1). Micrographs were screened manually to assess CTF quality and selected based on the presence of high-resolution Thon rings at least to 4 Å and beyond for further processing. A set of 10 randomly selected micrographs was used for manual picking of particles. Five 2D classes corresponding to the side views were then used for automated particle picking from the entire data set. About 520,000 particle images were picked from ~3500 micrographs using Relion 2.0[48]. Correction of the contrast transfer function was done by phase flipping during the following processing.

Initial steps of processing were done in Relion 2.0. The data set of particle images was subjected to 3 rounds of 2D classification to remove atypical images that included some ice contamination, particles that were overlapping or attached to the frame boundaries. The best classes representing mostly side views (~10,000 images) were selected firstly for the initial alignments and reconstruction. The portal protein gp6 map (EMD-1021)[10] low pass filtered to 20 Å, that revealed symmetry C12, was used as an initial model. Then we have performed a 3D classification using 5 classes. No symmetry was applied during these steps of analysis. Two classes were not well defined, two classes were rather similar with apparent C12 symmetry and the last one had mirrored structure with approximate 12-fold symmetry.

The two structures with the correct handedness, assessed by fitting the main domains of one subunit from the gp6 13 mer X-ray structure (PDB 2JES)[12], were averaged and used as a starting model to obtain the structure of the connector complex using the entire data set of 520,000 particles. Several iterations of 3D refinement alternated with 3D classification resulted in a structure at ~3.5 Å resolution.

In the next steps of refinement, we used a subset of movie frames from 2 to 21 after their alignment to eliminate frames with high dose of radiation and re-extracted particle images. These images and the model obtained in Relion 2.0 were exported in CryoSPARC[49] for further processing. After three rounds of 2D classification in CryoSPARC, 401,807 particle images were selected for the next two rounds of refinement. The structure of the gp6gp15gp16 complex with imposed symmetry C12 indicated that density of the region corresponding to the gp16 protein was nearly twice lower when compared to the densities of gp6 and gp15. This finding indicated that protein components within the connector have different symmetries. Therefore, we reduced the symmetry to investigate the distribution of densities corresponding to each protein component. Utilisation of 3-fold symmetry revealed six subunits in the region of gp16 implying that the gp16 ring has C6 symmetry. This symmetry was used at the final refinement of the complex structure. The cryo-electron densities corresponding to each of the three proteins were within the same range in the structure obtained.

The final structure of the complex was obtained at a resolution of 2.4 Å as estimated by the gold-standard Fourier shell correlation (FSC) at a 0.143 threshold and 2.7 Å at a 0.5 threshold. Local resolution

variations in the reconstruction was estimated using ResMap[50] (Supplementary Fig. 2d).

## Polypeptide chains tracing

Interpretation of the cryoEM map was done using Chimera[51], COOT[52], and PHENIX[53]. The initial atomic model building for gp6 was done by rigid body fitting of the stem domain (helices α3 and α5, residues 256–280 and 327–368 from the X-ray structure of a gp6 13-mer ring, PDB 2JES[12]). The remaining amino acids of the sequence were fitted de novo by manual tracing using COOT with extensive adjustments of the model to the cryoEM densities. That step was followed by real-space refinement of the atomic model against the experimental map in PHENIX, including restraints for rotamers, c-beta, and Ramachandran constraints. Note that the short β-strand β10 (residues 215–217) is only found in chains A, C, E, G, I, and K due to poor local cryo-electron density in the gp6 wing. The atomic models of gp15 and gp16 were build de novo in the cryoEM map. The amino acid sequences were traced manually with COOT and real space refinement with PHENIX in the cryoEM density map.

Refined atomic models of each protein in the connector complex were obtained by several rounds of rebuilding in COOT and real space refinement with rotamer, c-beta, and Ramachandran restraints in PHENIX. Manual inspection of the residues in the complete gp6, gp15, and gp16 polypeptide chains tracing was done using COOT to correct the orientation of rotamers and to trace regions of lower cryo-electron density quality (gp6 residues 170–176 and 217–241). Protein residues of the final atomic model show well-refined geometrical parameters: most favoured regions, 91.71%; additionally, allowed regions, 8.18%; and 0.10% of outliers in Ramachandran plots (Supplementary Table 1). Inter-subunit bonding, interaction interfaces, and subunits dissociation energy in the connector complex were computed using PISA[30] (Supplementary Tables 3 and 4). The protein interfaces electrostatic potential was calculated and displayed using Pymol[54].

## 3D homology search

Structure homology searches were carried with DALI[55] using individual subunits of the connector (this work) and the NMR structures of gp15 and gp16[19]. Pairwise structure comparison between domains of gp6 in the connector (this work) and in the assembly-naïve gp6 (PDB 2JES[12]) were carried with the DALI server[55] using default settings. The PDB coordinates of the corresponding domains were provided as input. The Cα coordinates of the polypeptide chain in the domains were used for calculation of the RMSD.

## Structure display

Figures were prepared using Pymol[54] and Chimera[51] software.

## Reporting summary

Further information on research design is available in the Nature Portfolio Reporting Summary linked to this article.

## Data availability

The data that support this study are available from the corresponding authors upon reasonable request. Coordinates of atomic models have been deposited in the Protein Data Bank under the accession code 7Z4W (gp6₁₂ - gp15₁₂ - gp16₆ complex) and the corresponding electron microscopy density map has been deposited in the Electron Microscopy Data Bank under the accession code EMD-14509 (gp6₁₂ - gp15₁₂ - gp16₆ complex).

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

## Acknowledgements

We thank Dr. D. Houldershaw and Y. Goudetsidis for computer support at Birkbeck throughout the duration of the project. Cryo-EM samples were prepared and optimised at the ISMB EM facility at Birkbeck College, with financial support from Wellcome Trust (079605/2/06/2). We acknowledge Diamond Light Source for access and support of the cryo-EM facilities at the UK National electron Bio-Imaging Centre (eBIC, under proposal EM14704) funded by the Wellcome Trust, the Medical Research Council UK, and the Biotechnology and Biological Sciences Research Council. We are grateful to Dr. Y. Chaban for his assistance with the data collection at the eBIC. We thank Dr. A. Isidro (VMS, CNRS, Gif-sur-Yvette, France) for a sample of HSV-1 C-capsids and Dr. R. Lurz (MPIMG, Berlin, Germany) for kindly providing the micrographs shown in Fig. 6. Part of this work was supported by institutional funding from CNRS.

## Author contributions

E.V.O. and P.T. designed and supervised the research. S.B. purified SPP1 DNA-filled capsids and connectors. N.L. prepared the EM grids, I.O., E.V.O. analyzed the EM data, and did structural analysis. I.O., E.V.O., S.R., and M-C.V. analyzed the map and performed modelling. P.T., E.V.O., and S.R. wrote the manuscript; I.O., E.V.O. S.R., and M.-C.V. have prepared the figures. All authors participated in the discussions of the results, figures, and critical reading of the manuscript. All authors approved the final version of the manuscript.

## Competing interests

The authors declare no competing interests.

## Ethics & Inclusion

Animals, human participants or their tissues were not involved or used in this study. No clinical trials were included.
