## [Peer Review File · Nature Communications]

CryoEM Structure and Assembly Mechanism of a Bacterial Virus Genome GatekeeperREVIEWER COMMENTS

Reviewer #1 (Remarks to the Author):

NCOMMS-22-24290-T

Title: CryoEM Structure and Assembly Mechanism of a Bacterial Virus Genome Gatekeeper

Authors: Igor Orlov, Stéphane Roche, Sandrine Brasiles, Natalya Lukoyanova, Marie-Christine Vaney, Paulo Tavares, Elena V Orlova

This manuscript describes the 2.7 Å cryoEM structure of the bacteriophage SPP1 portal and plug proteins after DNA packaging. The authors release the complex, which is composed of the portal (gp6), the first head completion protein (adapter, gp15), and the stopper protein (gp16), from DNA filled heads. Choosing this complex for this study simplifies the analysis as it does not have the entire DNA filled head with coat proteins that surround the portal or the tail attached to complicate the image processing. What is particularly lovely about this study is that there is a crystal structure of gp6, and NMR structures of gp15 and 16. These allow the authors to develop models through which they propose mechanisms for the sequential assembly of the complex by comparing the assembly naïve structures to those assembled in the complex. Portal protein is assembled within the head as a 12-mer but gp15 and 16 assemble as monomers onto the portal protein template. The authors describe the conformational changes that occur during assembly of the complex by comparing the crystal or NMR structures with those of the isolated complex, reported here. This is a well-written manuscript with beautiful figures. I particularly enjoyed figures 5 and 6, which move the manuscript from the reporting of a new structure to some speculation about how the assembly occurs and the evolution of these complex machines. I have just a few comments in descending order of importance:

1. I am surprised that there were no asymmetries noted in the structures as these are evident in other phage portal:adapter:tail assemblies. Did the authors do a final symmetry expansion to look for these? This analysis could be quite interesting and worth the effort.
2. Adding a sentence or two to the introduction about why the authors chose to solve this structure released from the DNA-filled heads would be helpful.
3. In figure 6, the way the figure is drawn seems to imply that the portal:adapter structures evolve from the herpes portal. I would suggest that it might be the other direction. If that is not what the authors intended to imply, they might consider redrawing the figure.
4. The authors use the portal structure to understand the effects of the many mutants made in SPP1 portal but do not discuss the mutants found in the wing domain.
5. In the paragraph starting on line 106, it is very hard to follow which of portal structures the authors are referring when comparing the different features.

Reviewer #2 (Remarks to the Author):

Orlov et al. describe the structure and assembly mechanism of the bacteriophage SPP1 portal gatekeeper, composed of gp6, gp15 and gp16. Comparison of this native structure derived by cryoEM to existing models of the individual components allowed them to suggest a mechanism for the conformational changes leading to its assembly. This assembly mechanism has implications for other tailed prokaryotic tailed viruses-herpesvirus lineage, as highlighted by the authors in the manuscript. Although the structure of several prokaryotic portal (and associated packaging/structural) proteins have been determined (as also highlighted by the authors), the assembly mechanisms of the gatekeeper keeping the packed genome inside the virion has been unknown to date. The authors additionally provide an updated stoichiometry of the gatekeeper in relation to the number of copies of gp16 deviating from previous findings (12-mer instead of 13-mer). Whereas the paper is well written, one can always speculate on the interest of the findings to the broader scientific community. Nevertheless, a discussion is included on the relevance to eukaryotic viruses (herpes cytomegalovirus), the data is in general very clearly presented and I would like to acknowledge the authors for including the clear supplemental tables both summarizing the known mutations in the SPP1 gatekeeper proteins but also highlighting the points of interaction between the gatekeeper components. The main strength of the paper lies in the detailed discussion on the assembly process (when contrasted to individual components) and to the comparison to other viruses.

I do have a few minor remarks, that I feel that the authors should clarify.

On line 96-98, the authors state that gp15 and gp16 are monomeric before assembly, and that assembly involves gp6-gp15, gp15-gp15, gp15-gp16 and gp16-gp16 interaction events. Here, a few more lines of interpretation would be useful, along with a reference to the extended video provided by the authors showing the assembly of initially gp15 monomers onto the dodecameric gp6 portal and subsequent association of gp16 to gp15.

On lines 202-210 the authors discuss the association of gp6 and gp15. As I read the paper, the assumption of that the dissociation energy between gp6 and gp15 is low, and gp15-gp15 interactions are required to stably assemble the complex. Do the authors have any experimental evidence (other than calculated dissociation energy) to support the fact that the gp15-gp15 interactions are required to keep the gp15-gp6 complex assembled, or is this finding merely speculative? The same holds true for the gp16-gp16 weak interactions (on lines 253-256), and the speculation that the gp16 hexamer is held primarily by bonds to the gp15 subunits. What would the benefits be of such weak interactions between gp6-gp15 and gp16-gp16 be on the function of the gatekeeper or the phage in general?

On lines 132-133 the authors write that they discern several groups of residues critical for DNA packaging (without indicating references). As I read the manuscript, these residues have been mapped previously and are here merely mapped onto the current structure(s). This should be clarified.

Reviewer #3 (Remarks to the Author):

The manuscript by Orlov et al. determines the high-resolution structure of the bacteriophage SPP1 portal protein in a complex with head completion proteins gp15 and g16 and is responsible for the retention of the DNA in the capsid. The work presented here revises the study carried out by the PI earlier (Orlova et al., EMBO J. 2003). Thanks to the development of cryo-EM technology over the last decade, the authors are now able to describe the complex responsible for DNA retention and the mechanism of phage head closure with a higher level of detail. The high-resolution structure also allows for structural interpretation of various mutations which were earlier described to interfere with the DNA packaging.

Major comment: The gatekeeper complex described here is in the contact with the phage capsid. It should be confirmed that the structure of the gatekeeper complex in the phage does not confront the interpretation of the data collected on the purified gatekeeper complexes. The throughput of current cryo-EM data acquisition should afford that even for larger phage heads such as SPP1.

Minor comments:

- Map to model FSC should be added to Extended Figure 2.

- The term 'electron density' should be replaced by 'cryo-EM density' throughout the manuscript.

The methodology section is well-written and contains a sufficient level of details.

NCOMMS-22-24290-T

Title: CryoEM Structure and Assembly Mechanism of a Bacterial Virus Genome Gatekeeper

Authors: Igor Orlov, Stéphane Roche, Sandrine Brasiles, Natalya Lukoyanova, Marie-Christine Vaney, Paulo Tavares, Elena V Orlova

Point-by-point answer to reviewers' comments:

We thank all reviewers for their high assessment of our work and the valuable comments on the work and manuscript. That would help to increase the impact of our manuscript. Please find below our answers where we address the raised questions in detail.

Reviewer #1 (Remarks to the Author):

NCOMMS-22-24290-T

Title: CryoEM Structure and Assembly Mechanism of a Bacterial Virus Genome Gatekeeper

Authors: Igor Orlov, Stéphane Roche, Sandrine Brasiles, Natalya Lukoyanova, Marie-Christine Vaney, Paulo Tavares, Elena V Orlova

This manuscript describes the 2.7 Å cryoEM structure of the bacteriophage SPP1 portal and plug proteins after DNA packaging. The authors release the complex, which is composed of the portal (gp6), the first head completion protein (adapter, gp15), and the stopper protein (gp16), from DNA filled heads. Choosing this complex for this study simplifies the analysis as it does not have the entire DNA filled head with coat proteins that surround the portal or the tail attached to complicate the image processing. What is particularly lovely about this study is that there is a crystal structure of gp6, and NMR structures of gp15 and 16. These allow the authors to develop models through which they propose mechanisms for the sequential assembly of the complex by comparing the assembly naïve structures to those assembled in the complex. Portal protein is assembled within the head as a 12-mer but gp15 and 16 assemble as monomers onto the portal protein template. The authors describe the conformational changes that occur during assembly of the complex by comparing the crystal or NMR structures with those of the isolated complex, reported here. This is a well-written manuscript with beautiful figures. I particularly enjoyed figures 5 and 6, which move the manuscript from the reporting of a new structure to some speculation about how the assembly occurs and the evolution of these complex machines. I have just a few comments in descending order of importance:

We thank the reviewer for the interest on our research and useful comments on the work and manuscript.

Reviewer 1

R1Q1 (Query 1). I am surprised that there were no asymmetries noted in the structures as these are evident in other phage portal:adapter:tail assemblies. Did the authors do a final symmetry expansion to look for these? This analysis could be quite interesting and worth the effort.

R1 Answer1. R1A1 We have done the asymmetric reconstruction of the SPP1 connector. However, the resolution of the asymmetrical structure was rather poor and did not allow identifying reliably features within the connector structure as, for example, in the portal flexible tunnel loops or in points of interaction between protein components. A high-resolution asymmetric structure would need a much larger dataset of connector particles. On the other hand, symmetry relaxation from an initial 12-fold symmetry to 6-fold symmetry of the connector complex was necessary to resolve the 6-fold symmetric gp16 ring structure and its interactions with gp15 (see Methods). This symmetry transition enables symmetry matching of the connector gp16 with the 6-fold symmetric tail. We are sure that the absence of other significant asymmetry in the connector was one of the reasons why we achieved a high-resolution 3D reconstruction of the highly stable connector complex.

Asymmetry was previously reported for complexes where the portal interacts with the 5-fold capsid vertex, helical DNA or core/ejection proteins (e.g. reviewed in Dedeo et al 2019; ref 5 of the manuscript). These analyses of interactions between portal, capsid proteins, and DNA were carried out with the portal embedded in the capsid structure.

It would be interesting to make such type of study for the SPP1 connector within the SPP1 capsid. However, analysis of the SPP1 DNA-filled capsid with symmetry relaxation is a demanding task to reveal all interactions between connector components and capsid proteins requiring a resolution that will enable us to see all amino acids. We thus feel that such study can build a story on its own (see also answer R3A1 to referee 3 below). Our high-resolution structure presented in this manuscript of the connector purified from DNA-filled capsids focus on the interactions between the 3 connector protein components, that are rather flexible and do not interact with each other in solution before assembly. The main novelty of our work based on the atomic model of cryoEM structure is uncovering the structural organization of the connector complex, the conformational changes leading to its assembly, and explaining why this complex is so stable at the post-DNA packaging state. These mechanisms and their evolutionary implications are novel.

R1Q2. Adding a sentence or two to the introduction about why the authors chose to solve this structure released from the DNA-filled heads would be helpful.

R1A2: We thank the reviewer for this useful suggestion. We have now changed the text of Introduction (labelled in magenta in the last version of the manuscript, page 3, bottom paragraph) to clarify this point:

“The connector was extracted from tailless DNA-filled capsids to avoid capsid protein subunits surrounding it at mismatched positions. This strategy allowed us to focus image processing on analysis of the connector complex and achieve a high-resolution structure”

R1Q3. In figure 6, the way the figure is drawn seems to imply that the portal adapter structures evolve from the herpes portal. I would suggest that it might be the other direction. If that is not what the authors intended to imply, they might consider redrawing the figure.

R1A3: We thank the reviewer for this comment. The previous version of Figure 6 did not imply that phage portals evolved from the herpes portal. The structure shown on the top of the figure is the SPP1 portal protein 12-mer given as an example of the portal system from a primordial prokaryotic tailed virus. The comparison of portal structures supports the notion that the portals from tailed viruses and herpesviruses descend from a common protein ancestor. This ancestor was conceivably present in an ancient prokaryotic virus, as prokaryotes precede eukaryotes in the evolution of Life. Our hypothesis is that the portal of an ancestral tailed virus acquired the turret domain and lost concomitantly the interaction with the adaptor to originate the herpesviruses tailless virions. Figure 6 and the Discussion text (page 13, last paragraph of Discussion, labelled in magenta in the last version of the manuscript) were modified accordingly:

“Prokaryotes preceded eukaryotes in evolution. We thus hypothesize that an ancestral prokaryotic tailed virus evolved by insertion of the turret domain in the portal clip and loss of interaction with the adaptor protein to originate the portal ancestor of eukaryotic herpesviruses virions (Fig. 6). Collectively, our connector structure and comparative structural biology study uncover the molecular mechanism of assembly of the complete viral DNA gatekeeper, providing a long-awaited molecular framework to trace the evolutionary divergence path between herpesviruses and different tailed phage families.”

R1Q4. The authors use the portal structure to understand the effects of the many mutants made in SPP1 portal but do not discuss the mutants found in the wing domain.

R1A4: The reviewer is right. Mutations in the wing domain that affect DNA packaging were originally only shown in Fig. 2e and described in Supplementary Table 2. They are expected to cause local destabilisation of the wing domain structure but the way they affect DNA packaging is difficult to interpret. This is now stated in a dedicated main text paragraph of the revised manuscript (page 5, second paragraph from the bottom, labelled in magenta in the last version of the manuscript).

“Several mutations mapped in the portal wing (Fig. 2e, in orange) apparently cause local destabilization of this domain (Supplementary Table 2). It is difficult to interpret how their effect on the structure disrupts DNA packaging through the portal central tunnel.”

R1Q5. In the paragraph starting on line 106, it is very hard to follow which of portal structures the authors are referring when comparing the different features.

R1A5: We have rephrased the text of this paragraph to describe more distinctly the differences between the portal 12mer structure within the connector and the assembly-naïve 13mer portal when comparing these structural elements (pages 4-5, last

paragraph in page 4 that continue in the page 5, in the top of the page, labelled in the last version of the manuscript). We hope it makes the comparison clearer:

“The gp6 12-mer within the connector and the assembly-naïve gp6 13-mer are very similar in the inner part of the wing domain (residues 29-54, 88-169, 178-208 and 369-420, with RMSD between the C α positions of 0.7 Å) and in the portal stem (residues 55-62, 81-87, 256-280, 327-346, RMSD 0.5 Å) (Fig. 2b). However, noticeable structural differences were found within the crown area, tunnel loops, β -hairpins, and clip domains (Fig. 2b). The tunnel loop forms a short helix in gp6 within the connector, while this helix is absent in the gp6 13-mer¹² (Fig. 2b). The low cryo-electron density of the tunnel loop in the connector gp6 suggests that this segment is rather flexible. The most significant difference between the connector gp6 and the gp6 13-mer is found in the clip domain that makes the most extensive intersubunit bonding in the two portal oligomers (Fig. 2c,d). The clip is formed by an intersubunit triple-stranded β -sheet, composed of strands β 12 and β 14 from subunit i and β 13 from subunit $i-1$, and helix α 4 from subunit i (Fig. 2c,d). These structural elements are found in both gp6 structures (12-mer and 13-mer) but exhibit visible shifts between residues Gln282 at the end of helix α 3 and Pro325 at the beginning of helix α 5 (Fig. 2b-d). In the gp6 13-mer, helix α 4 has a tilted orientation¹² with its N-terminus Pro296 residue localized at the clip bottom (Fig. 2c; Extended Data Fig. 3a). In the connector gp6, Pro296 is shifted upwards by 7 Å bringing helix α 4 to a more horizontal orientation closer to the gp6 β -hairpin of adjacent subunit $i+2$ (Fig. 2d). These changes in the clip domains of the connector gp6 lead to formation of pockets where the C-termini of gp15 bind to the portal oligomer (Extended Data Fig. 3b,c). Such pockets are absent in the assembly-naïve gp6 13-mer (Extended Data Fig. 3a) explaining why it does not bind gp15²⁴.”

Reviewer #2 (Remarks to the Author):

Orlov et al. describe the structure and assembly mechanism of the bacteriophage SPP1 portal gatekeeper, composed of gp6, gp15 and gp16. Comparison of this native structure derived by cryoEM to existing models of the individual components allowed them to suggest a mechanism for the conformational changes leading to its assembly. This assembly mechanism has implications for other tailed prokaryotic tailed viruses-herpesvirus lineage, as highlighted by the authors in the manuscript. Although the structure of several prokaryotic portal (and associated packaging/structural) proteins have been determined (as also highlighted by the authors), the assembly mechanisms of the gatekeeper keeping the packed genome inside the virion has been unknown to date. The authors additionally provide an updated stoichiometry of the gatekeeper in relation to the number of copies of gp16 deviating from previous findings (12-mer instead of 13-mer). Whereas the paper is well written, one can always speculate on the interest of the findings to the broader scientific community. Nevertheless, a discussion is included on the relevance to eukaryotic viruses (herpes cytomegalovirus), the data is in general very clearly presented and I would like to acknowledge the authors for including the clear supplemental tables both summarizing the known mutations in the SPP1 gatekeeper proteins but also highlighting the points of interaction between the gatekeeper components. The main strength of the paper

lies in the detailed discussion on the assembly process (when contrasted to individual components) and to the comparison to other viruses.

We thank the reviewer for the positive evaluation of our work and the manuscript. We appreciate her/his useful comments that are addressed below.

I do have a few minor remarks, that I feel that the authors should clarify.

R2Q1. On line 96-98, the authors state that gp15 and gp16 are monomeric before assembly, and that assembly involves gp6-gp15, gp15-gp15, gp15-gp16 and gp16-gp16 interaction events. Here, a few more lines of interpretation would be useful, along with a reference to the extended video provided by the authors showing the assembly of initially gp15 monomers onto the dodecameric gp6 portal and subsequent association of gp16 to gp15.

R2A1: We provide a more comprehensive description of the sequential interactions that occur during connector assembly in the section of Results suggested by the reviewer (page 4, The end of the first paragraph in the Results. Labelled in magenta in the last version of the manuscript).

" Gp15 and gp16 are monomeric before they assemble in the connector²⁴. After DNA packaging, gp15 binds to the portal forming a dodecameric cyclical oligomer. This assembly step occurs independently of gp16²⁰. Closure of the connector complex requires binding of gp16 to gp15, leading to assembly of the gp16 6-mer (Fig. 1) that retains phage DNA inside the viral capsid^{10,20}. These observations combined with the connector structure obtained (Fig. 1) reveal that connector assembling involves gp6-gp15, gp15-gp15, gp15-gp16, and gp16-gp16 interaction events."

Nonetheless, it is important to mention that the structural analyses described in the following sections of Results delivers the basics for molecular modelling of the program of conformational changes in connector proteins during viral assembly. These steps are demonstrated by the model in Fig. 5 and in the Extended Movie 1 that we find more suitable to refer only in the manuscript Discussion.

R2Q2. On lines 202-210 the authors discuss the association of gp6 and gp15. As I read the paper, the assumption of that the dissociation energy between gp6 and gp15 is low, and gp15-gp15 interactions are required to stably assemble the complex. Do the authors have any experimental evidence (other than calculated dissociation energy) to support the fact that the gp15-gp15 interactions are required to keep the gp15-gp6 complex assembled, or is this finding merely speculative? The same holds true for the gp16-gp16 weak interactions, and the speculation that the gp16 hexamer is held primarily by bonds to the gp15 subunits. What would the benefits be of such weak interactions between gp6-gp15 and gp16-gp16 be on the function of the gatekeeper or the phage in general?

R2A2: Dissociation energies were calculated only computationally using PISA (see Methods).

Experimental measurement of association-dissociation constants of these complexes is very difficult because the portal is primed for gp15 binding only after DNA packaging. Such measurements would thus require either (i) a high yield *in vitro* system for coupled DNA packaging-connector assembly, or (ii) purification of large amounts of biochemically homogeneous capsids that packaged DNA for interaction with gp15 (and subsequently gp16). We did not succeed to achieve so far any of these experimental setups.

Therefore, our assessment of the gp6-gp15, gp15-gp15, gp15-gp16, and gp16-gp16 interaction stability was based only on the evaluation of the interfaces by PISA, including the dissociation energies of the interfaces, the number of interchain hydrogen bonds and salt bridges, and the surfaces of interaction (Supplementary Table 4). These criteria are commonly used to assess stability of protein complexes (see e.g. reference 31). For the revised version of the manuscript, we made a more detailed analysis of the interfaces using PISA that led to some minor corrections of interface areas and assessments of dissociation energies (Supplementary Table 4).

Given our model of sequential assembly, we consider that the gp15 monomer binds to the gp6 ring. This interaction represents a calculated dissociation energy of -1.6 kcal/mol for a surface of interaction of ~700 Å². Comparatively the gp15-gp15 intersubunit interface represents a calculated dissociation energy of -13.3 kcal/mol for a surface of interaction of ~1245 Å² (Supplementary Table 4). Although these numbers are obtained from computational analyses of intermolecular interfaces, their difference provides a good indication that binding of gp15 to the gp6 ring is weak and that the subsequent homo-oligomerization of gp15 establishing gp15-gp15 bonding stabilizes the structure. These issues were further elaborated in the revised manuscript (page 8, second paragraph from the top, labelled in magenta in the last version of the manuscript):

“In contrast, adjacent gp15 subunits have an interaction surface of ~1250 Å² and a dissociation energy of -13.3 kcal/mol (Supplementary Table 4). Furthermore, each gp15 subunit establishes interfaces with two adjacent gp15 subunits as its cyclical 12mer forms during connector assembly. The initial attachment of gp15 monomers to gp6 is thus followed by gp15 intersubunit interactions that ensure stable association of gp15 to the gp6 oligomer.”

Binding of a gp16 subunit to the gp15 ring represents a calculated dissociation energy of -11.3 kcal/mol for a surface of interaction of ~1040 Å². Comparatively, the gp16-gp16 interface represents a calculated dissociation energy of -1.9 kcal/mol for a surface of interaction of ~510 Å² (Supplementary Table 4). Furthermore, there are only 7 intersubunit intermolecular hydrogen bonds at the gp16-gp16 interface. Five of those are found in the region of gp16 that closes the connector tunnel (Fig. 4b) while almost no gp16-gp16 bonding is found within the outer region of the gp16 hexamer. Collectively, these observations support our assertion that the gp16 hexamer is stabilized primarily by its interactions with the gp15 ring. We have rephrased the sentence (Page 9, bottom paragraph, labelled in magenta in the last version of the manuscript) to document better this point:

“The interaction between gp16 subunits is notably weak with a surface of 510 Å² between adjacent subunits and a calculated dissociation energy of -2 kcal/mol (Supplementary Table 4), suggesting that bonds to gp15 subunits play a significant role to hold the gp16 hexamer in the connector.”

The advantage of such weak gp6-gp15 and gp16-gp16 interactions is an interesting question. We can only speculate at present:

- There might be an advantage that gp15 subunits bind to gp6 weakly in order to avoid that gp15 interferes with terminase binding to the portal during DNA packaging, a reaction that precedes gp15 binding (note that the terminase and gp15 bind to the same interface of gp6 (see the first paragraph in page 7, main manuscript).
- The small interface area and the low dissociation energy between gp16 subunits might be a requirement for the conformational changes that lead to opening of the gp16 central channel for DNA passage to the tail tube.

R2Q3. On lines 132-133 the authors write that they discern several groups of residues critical for DNA packaging (without indicating references). As I read the manuscript, these residues have been mapped previously and are here merely mapped onto the current structure(s). This should be clarified.

R2A3: The paragraph including lines 132-133 of the original manuscript submission starts by stating that the structure “... allows to rationalize the effect of mutations previously shown to impair the SPP1 DNA packaging process^{25,26}”, citing the original references 25,26. Those references were also provided in Supplementary Table 2 that includes a description of the mutation phenotype. In the revised manuscript, those references are now cited also in the closing sentence of the the first paragraph in the Results section on the Mapping of vital DNA packaging mutations in the portal structure, page 5 of the revised manuscript).

“Residues critical for DNA packaging^{25,26} are located in different structural elements of the portal structure (Fig. 2e)”

Reviewer #3 (Remarks to the Author):

The manuscript by Orlov et al. determines the high-resolution structure of the bacteriophage SPP1 portal protein in a complex with head completion proteins gp15 and g16 and is responsible for the retention of the DNA in the capsid. The work presented here revises the study carried out by the PI earlier (Orlova et al., EMBO J. 2003). Thanks to the development of cryo-EM technology over the last decade, the authors are now able to describe the complex responsible for DNA retention and the mechanism of phage head closure with a higher level of detail. The high-resolution structure also allows for structural interpretation of various mutations which were earlier described to interfere with the DNA packaging.

R3Q1. Major comment: The gatekeeper complex described here is in the contact with the phage capsid. It should be confirmed that the structure of the gatekeeper complex in the phage does not confront the interpretation of the data collected on the purified gatekeeper complexes. The throughput of current cryo-EM data acquisition should afford that even for larger phage heads such as SPP1.

R3A1. The structure of the complete SPP1 capsid will be of course of great interest. However, it is a challenging task to produce an asymmetric cryoEM reconstruction of the SPP1 DNA-filled capsid to determine the structure of the connector at a resolution comparable to the one reported here for the isolated complex (see our answer to the first reviewer **R1A1**). That would be necessary for a rigorous structural comparison as the one suggested by the reviewer. In the present study, high-resolution cryoEM was critical for non-ambiguous tracing of the complete set of polypeptide chains in the connector. Absence of the symmetrically mismatched capsid protein around the connector was instrumental to reach this high resolution, as indicated by reviewer 1 and now described in Introduction (labelled in magenta in the last version of the manuscript, page 3, bottom paragraph). Thus, we find that the cryo-EM structure of the connector purified from DNA-filled capsids at the post-DNA packaging assembly state was methodologically the more suitable avenue to address the questions raised on this study.

The novelty of our work is related substantially to the quality of the EM connector structure followed by building the atomic models of its three proteins. That allowed to uncover the connector structural organization and to suggest a program of conformational changes of the protein components at the molecular level explaining the sequence of assembly reactions of the connector. These mechanisms and their evolutionary implications are novel. Therefore, we feel that our work in the present form builds a complete study of broad interest. The asymmetric reconstruction of the SPP1 DNA-filled capsid suggested by the reviewer fits best to another study addressing additional questions related to the portal-capsid interaction, distortion of the capsid lattice and DNA organization at the portal vertex and others.

Minor comments:

R3Q2. - Map to model FSC should be added to Extended Figure 2.

R3A2. The information requested is now included in modified Extended Data Figure 2

R3Q3. - The term 'electron density' should be replaced by 'cryo-EM density' throughout the manuscript.

R3A3. We modified the manuscript text following the reviewer's suggestion

The methodology section is well-written and contains a sufficient level of details.

We are pleased that the reviewer found our methods section complete and comprehensive

We have done some *additional modifications to improve clarity and/or accuracy of figures and supplementary tables*:

Figures 1d and 4a: The nomenclature of strands β_6 and β_6' of gp16 in the connector was changed to β_7 and β_7' in order to follow the original nomenclature used for β_7 in the gp16 monomer structure of reference 19.

Figure 6: figure was modified as suggested by reviewer 1. See **R1A3**.

Extended Data Fig. 3: the structures shown are now identified in each figure panel header

Extended Data Fig. 4: Residue D314 and strand β_{13} of gp6 are labelled for completeness

Supplementary Table 2: The Table was modified to provide more detail on the description and effect of some gp6 single amino acid substitutions that impair DNA packaging.

Supplementary Table 3: The Table was expanded to improve clarity. We added two columns to the Table to show the residues location in structural elements of the connector proteins. Minor corrections were made in other columns.

Supplementary Table 4: This Table was modified to improve identification of the interfaces that are analysed and to include buried surface areas at each interface. The different interactions of adjacent gp15 subunits ($j-1$ and j) with gp16 (see **Fig. 3f,g**) at the 12 (gp15) to 6-fold symmetry (gp16) transition region of the connector structure are presented separately.

REVIEWERS' COMMENTS

Reviewer #1 (Remarks to the Author):

The authors have carefully considered and fixed the issues I raised.

Reviewer #2 (Remarks to the Author):

Orlov et al. describe the structure and assembly mechanism of the bacteriophage SPP1 portal gatekeeper, composed of gp6, gp15 and gp16. The authors have addressed all my concerns mostly regarding the gp6, gp15 and gp16 assembly and interactions, and have revised the manuscript to include these clarifications. I have no further concerns regarding this well written manuscript.

Reviewer #3 (Remarks to the Author):

This reviewer still believes that it'd be highly beneficial if additional data (even if low-resolution) on the gate-keeper complex in the context of the bacteriophage structure are included. However, I also understand the potential technical difficulties.

Besides that, I have no other comments.

NCOMMS-22-24290-A

Title: CryoEM Structure and Assembly Mechanism of a Bacterial Virus Genome Gatekeeper

Authors: Igor Orlov, Stéphane Roche, Sandrine Brasiles, Natalya Lukoyanova, Marie-Christine Vaney, Paulo Tavares, Elena V Orlova

Point-by-point answer to reviewers' comments:

Reviewer #1 (Remarks to the Author):

The authors have carefully considered and fixed the issues I raised.

We thank the reviewer for the positive evaluation of our revised version of the manuscript

Reviewer #2 (Remarks to the Author):

Orlov et al. describe the structure and assembly mechanism of the bacteriophage SPP1 portal gatekeeper, composed of gp6, gp15 and gp16. The authors have addressed all my concerns mostly regarding the gp6, gp15 and gp16 assembly and interactions, and have revised the manuscript to include these clarifications. I have no further concerns regarding this well written manuscript.

We thank the reviewer for the positive evaluation of our revised version of the manuscript

Reviewer #3 (Remarks to the Author):

This reviewer still believes that it'd be highly beneficial if additional data (even if low-resolution) on the gate-keeper complex in the context of the bacteriophage structure are included. However, I also understand the potential technical difficulties.

Besides that, I have no other comments.

The structure of the connector within the complete SPP1 capsid will be of great interest. However, we still think that a rigorous comparison between the isolated connector and its structure embedded in the capsid would need high-resolution data for a unambiguous analysis at the molecular level. To that goal, it would be necessary to obtain an asymmetric high-resolution cryoEM reconstruction of the viral capsid. That is technically very challenging and such reconstruction could build a story on its own focusing on the interaction of connector components with the symmetrically mismatched capsid lattice. On the other side, the atomic structure reported in this work proved highly suitable to uncover the mechanism of connector assembly and provided novel insight on the evolution of tailed phages and herpesviruses. Both are novel and important findings of general interest.